# Optimization of Photobiomodulation Dose in Biological Tissue by Adjusting the Focal Point of Lens

Chuan-Tsung Su [1], Fu-Chien Chiu [2], Shih-Hsin Ma [3] and Jih-Huah Wu [4],*

1   Department of Healthcare Information and Management, Ming Chuan University, Taoyuan 33348, Taiwan; ctsu@mail.mcu.edu.tw
2   Department of Electronic Engineering, Ming Chung University, Taoyuan 33348, Taiwan; fcchiu@mail.mcu.edu.tw
3   Department of Photonics, Feng Chia University, Taichung 40724, Taiwan; shma@fcu.edu.tw
4   Department of Biomedical Engineering, Ming Chuan University, Taoyuan 33348, Taiwan
*   Correspondence: wujh@mail.mcu.edu.tw

**Abstract:** The optical power density in biotissue is an important issue for photobiomodulation (PBM) clinical applications. In our previous study, the maximal dose and the power density distributions of 830 nm lasers under human skin could be exactly calculated and measured. In this work, the laser power density in tissue can be changed by adjusting the focal point of the lens. From the experimental results, it is evident that the power densities on the attached gingiva and the surrounding tissues can be improved. Thus, the dose of a near-infrared (NIR) laser in the target tissue can be increased with a suitable lens. Most importantly, focusing lasers on deeper tissue can avoid any damage to the skin. This study provides a dose optimization method on the target tissue, and the results can be applied to clinical applications, especially laser acupuncture (LA).

**Keywords:** photobiomodulation; dose; near infrared (NIR); laser acupuncture (LA)

## 1. Introduction

Photobiomodulation (PBM) has been published in many clinical applications. Light wavelengths (nm), power density ($W/cm^2$), energy density ($J/cm^2$), operation frequency (Hz), spot size ($cm^2$), and penetration depth are key parameters for the PBM applications. In PBM, the dose is an important parameter for biological photoresponses. The dose of PBM is defined as the energy density ($J/cm^2$) received on the targeted area. Namely, the light dose is the product of power density and duration of time for medical applications. In our previous report, low back pain was reduced by laser acupuncture [1,2]. In a study of PBM use in animals [3], the pain factors were reduced by radiating the Zusanli (ST36) acupoint with laser acupuncture. The irradiation dose was a crucial determinant of PBM effectiveness in this study. On the other hand, a biphasic dose response was observed in brain wave stimulations [4,5]. An array of lasers were used to radiate the left palms of the subjects. Their brain waves were changed; in particular, the alpha band increased in some regions and the beta band decreased in open-eye conditions. The subjects were relaxed [4]. According to the light-emitting diode (LED) [5] and laser [4] test results, the higher the dose, the higher the alpha and theta activity, and there is a positive correlation between the dose and the alpha and theta activities. On the contrary, higher LED dosage induced the beta activity of the right parietal hemisphere [5], but lower laser dosage reduced the beta activity [4], and higher beta activity means a state of alertness or anxiety. According to the Arndt–Schulz Law, a low dose of PBM is beneficial, but a high dose inhibits physiological activity [6–8]. A suitable dose for the target can increase the efficiency of PBM clinical applications.

To understand the optical characteristics of PBM in biotissue, a quantitative analysis method for dose and penetration depth in the tissue is important. The backscattering light on the palm with a He-Ne laser received by a detector in different positions was investigated,

and a good correlation between the simulation and experimental results was found [9]. Two- and four-layer models of the skin were used to simulate light propagation, and the backscattering light received by a detector was published [10]. The optical penetration depths of light in human mucous tissue were performed using a commercially available spectrophotometer with an integrating sphere [11]. The spectral ranges for maximal penetration depths are 800–900 nm and 1000–1100 nm. The penetration depth showed in human ex vivo tissues with 835 nm lasers is higher than that of 632.8 nm, 635 nm, and 780 nm lasers [12]. In animal tissues, a minor attenuation on the rat skin, pig fat, and pig muscle with a 904 nm laser was measured [13]. Meanwhile, the pig skin showed slight attenuation from the 830 nm laser. The light penetrations depend on wavelength as well as the sample used in the experiment. The well-transmitted wavelength would be different for human ex vivo tissues [12] and animal tissues [13]. The use of optical fiber probes as a quantitative analysis method was designed for light dosimetry in bovine muscle [14] and chicken tissue samples [15]. For the power density of 1.0 mW/cm$^2$ measured in bovine tissue samples, the penetration depth of an 808 nm laser can be predicted to be 3.4 cm [16]. Recently, Lan and co-workers reported a prototype of the LA system in which the focus of a laser can be changed by lift–thrust operation [17]. However, the real dose and its distribution in deeper tissue are unknown when the laser radiates on the skin.

NIR light showed better penetration in soft tissues [18]. The 830 nm laser energy ranged from 36 J/cm$^2$ to 360 J/cm$^2$, and was used to evaluate the safety in rat models. The results revealed no macroscopic or microscopic change in the tissue architecture [19]. Based on the beneficial wavelength range [20], mechanisms of photorespose [21–26], higher penetration depth [12], and safety [19], the 830 nm laser is a suitable laser source for PBM applications. According to our recent study, a non-invasive sensing system was developed for laser power density measurement on the attached gingiva [27]. However, since the laser power density decays sharply around the peripheral tissues, this situation should be improved for increasing PBM efficacy. In this work, changing the focal points of 830 nm lasers in the tissue was proposed to increase the power distribution on the target and around the peripheral tissues.

## 2. Materials and Methods

In this work, the aluminum gallium indium phosphide diode laser (model: U-LD-66A051Ap/Dp, Pocket Laser, Union Optronics Corp., Taoyuan, Taiwan) with a wavelength at 660 nm ± 5 nm and the aluminium gallium arsenide diode laser (model: T8350, pocket laser, Opto Focus Co., Ltd., New Taipei City, Taiwan) with a wavelength at 830 ± 10 nm were used as light sources. Both lasers have the same output power of 30 mW. An auto current control circuit was designed for driving the laser diode. An aspheric glass lens was used in this experiment. High-speed sensitive detectors (model: PD15-22C, Silicon PIN Photodiode, Everlight Electronics Co., Ltd., New Taipei City, Taiwan) were used for the power density measurement of 660 nm and 830 nm lasers radiated on the targets [27]. Nine detectors were mounted on the retainer (manufactured by College of Oral Medicine, Chung Shan Medical University, Taiwan) which was placed on the attached gingiva. A flexible facial fixture appliance (manufactured by iFaceDesign Technology Inc., New Taipei City, Taiwan) provided a stable position for the laser source (Figure 1).

A male case is presented in this study. Five detectors were used to measure the backscattering light of 660 nm and 830 nm lasers on the orbicularis oris, forearm, back of the hand, and palm areas. In addition, the power density distributions of an 830 nm laser after penetrating the orbicular oris on the attached gingiva were measured by nine detectors. The attached gingiva was in the best position possible to measure the power density of NIR after the laser penetrated the orbicular oris. In this case, the thickness of the orbicularis oris was 1.0 cm.

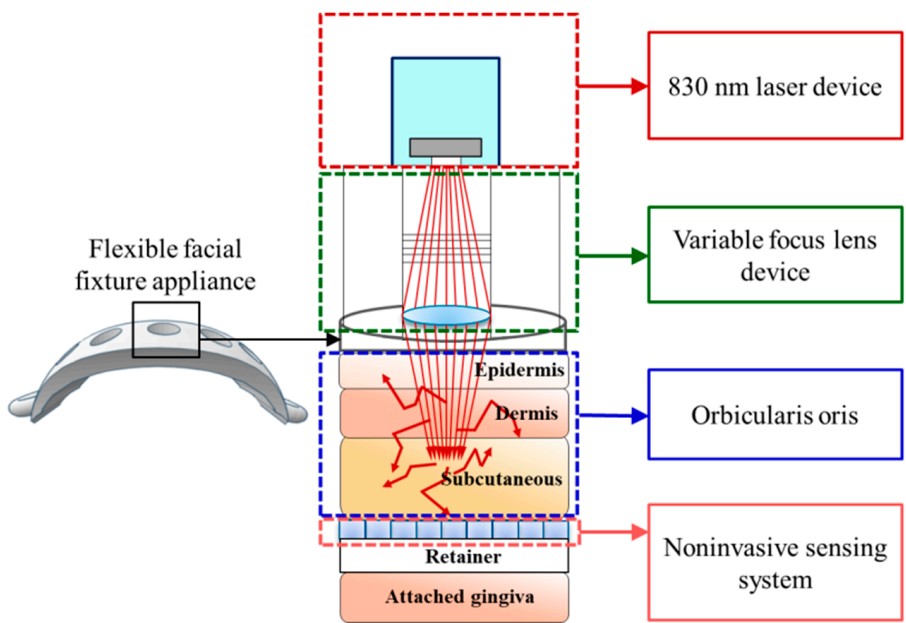

**Figure 1.** Schematic of the 830 nm laser radiated on the orbicularis oris by noninvasive sensing system.

The optical characteristics of backscattering on skin by 660 nm and 825 nm lasers were simulated. The Fresnel reflection coefficients [28,29] and Snell's law were applied for the optical properties simulation. The skin model of reflection parts was mentioned [30]. The incident power of the 660 nm and 825 nm lasers were set as 30 mW for the simulation. The epidermis layer (refractive index $n = 1.34$), dermis layer ($n = 1.4$), and subcutaneous layer ($n = 1.44$) were used to determine the reflection of the 660 nm laser in the tissue [31,32]. In the same way, the epidermis layer (refractive index $n = 1.34$), dermis layer ($n = 1.55$), and subcutaneous layer ($n = 1.45$) were used to determine the 825 nm laser reflection in the tissue [33,34]. The reflection intensity of the 660 nm and 825 nm lasers irradiated on the skin, with the incident angle ranging from 1° to 60°, was made by MATLAB 2015b according to the following formula [28,29]:

$$R(\theta) = \frac{1}{2}\left[\frac{sin^2(\theta - \theta_t)}{sin^2(\theta + \theta_t)} + \frac{tan^2(\theta - \theta_t)}{tan^2(\theta + \theta_t)}\right], \text{ if } 0 < \theta < sin^{-1}\left(\frac{n_{k+1}}{n_k}\right) \tag{1}$$

where $R$ is the Fresnel reflection coefficient, and $\theta$ and $\theta_t$ are the angles of the photon packet incidence on the layer boundary and transmittance, respectively. The refractive indexes of the medium layers are $n_k$ and $n_{k+1}$.

Furthermore, photon patterns were simulated after the 825 nm laser penetrated the skin by collimated and un-collimated lasers. The epidermis layer (refractive index $n = 1.34$; thickness: 0.065 mm), dermis layer ($n = 1.55$; thickness: 1.25 mm), and subcutaneous layer ($n = 1.45$; thickness: 12 mm) were used to determine the reflection of 825 nm lasers in the tissue. In addition, 10 million photon packages as light sources were adopted for assessing the light scattering in the tissue. The distance between the photon packages and the skin was 1 mm. Light scattering in the two-dimensional tissue model was performed by Advanced Systems Analysis Program according to the Monte Carlo of Henyey–Greenstein function [35,36]:

$$P_{HG}(\theta, g) = \frac{1 - g^2}{[1 + g^2 - 2g cos(\theta)]^{\frac{3}{2}}} \tag{2}$$

where $P_{HG}$ is the light scattering probability at the angle $\theta$ and $g$ is the anisotropic coefficient. During the simulation work, the effects of optical characteristics of blood flow and composition of the blood vessels were neglected.

## 3. Results

Based on our recent work [27], the dose and laser distributions of an 830 nm laser on the attached gingiva were measured with precision (Figure 2a). The power density analysis for collimated and uncollimated lasers on the upper attached gingiva of a male are shown in Figure 2b,e, respectively. Two-dimensional skin models were simulated by use of the Monte Carlo of Henyey–Greenstein function simulation (Figure 2c,f). When an 830 nm laser penetrates the skin, a more uniform pattern of light can be predicted for a collimated laser (Figure 2d,g). The simulated results indicate that collimated lasers can increase the power density on the surrounding targeted tissues.

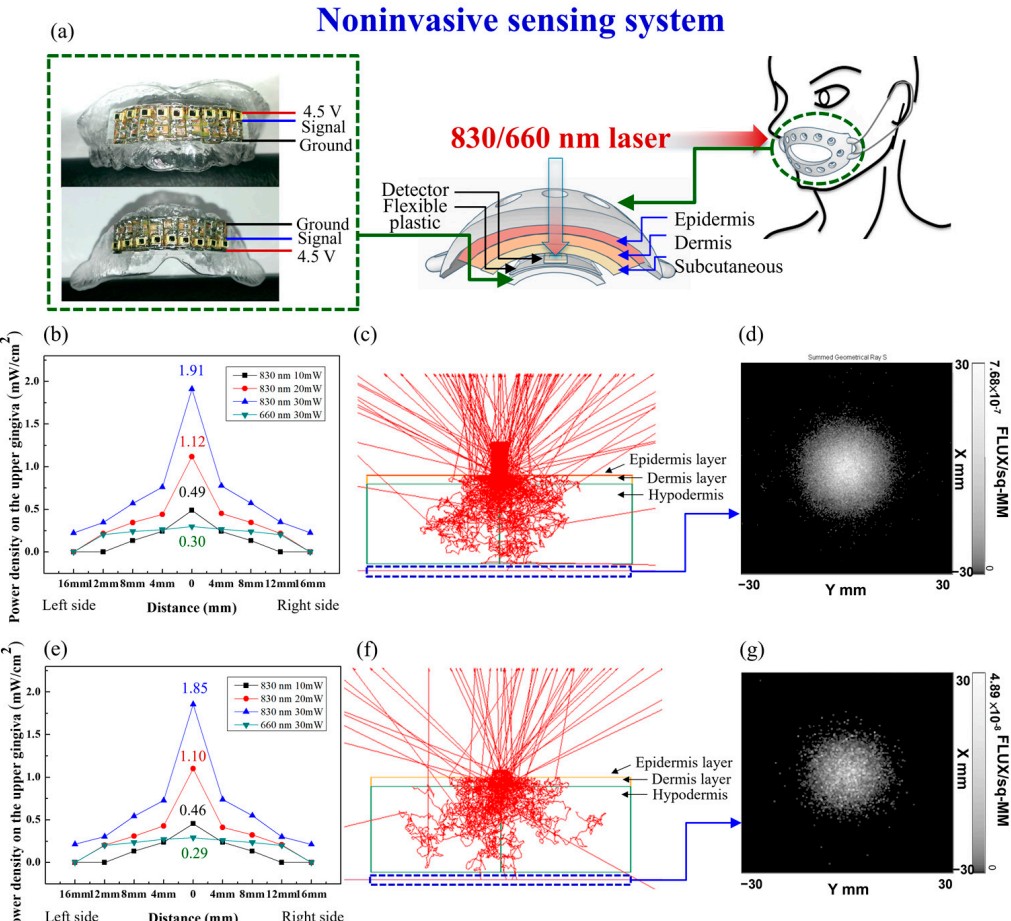

**Figure 2.** (**a**) Power densities of 830/660 nm lasers measured on the attached gingiva by a non-invasive sensing system. (**b**) Power densities analysis for a collimated laser on the upper attached gingiva for a male [27]. (**c**) The simulation of light scattering in a two-dimensional skin model and (**d**) the photon pattern after 825 nm laser penetrated skin for collimated laser. (**e**) Power density analysis for a laser without a lens on the upper attached gingiva for a male [27]. (**f**) The simulation of light scattering in a two-dimensional skin model and (**g**) the photon pattern after a 825 nm laser penetrated in skin for an uncollimated laser.

The reflection intensity of 660 nm and 825 nm lasers with incident angles ranging from 1° to 60° on the skin were simulated (Figure 3a,b). When the incidence angles of 660 nm and 825 lasers on the skin are increased, higher reflection intensity can be predicted. The simulated results indicate that the power density of the lasers decreases at a high incidence angle. Therefore, the light penetration of an 825 nm collimated laser in a two-dimensional skin model is more concentrated compared to that without a lens (Figure 2c,f).

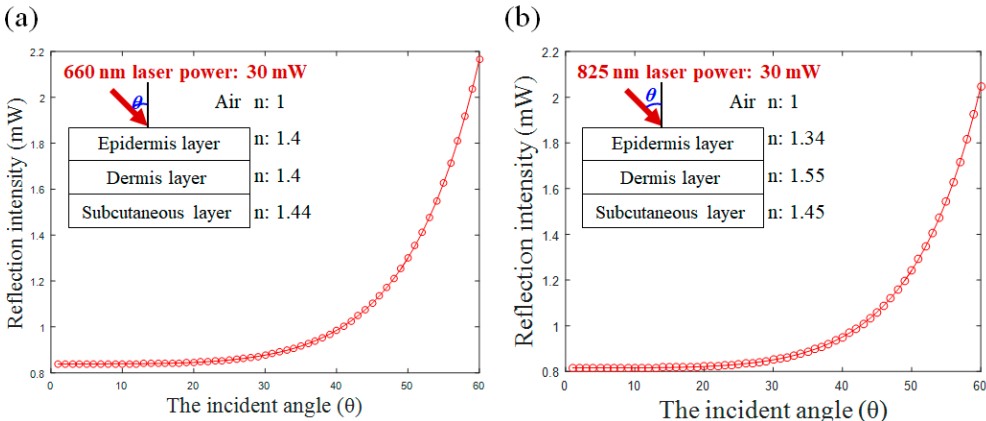

**Figure 3.** The simulation of (**a**) 660 nm and (**b**) 825 nm lasers irradiated on skin with different incident angles. Here, *n* is the refractive index.

The backscattering light of 660 nm and 830 nm lasers in different skin areas were measured (Figure 4). The skin areas included the orbicularis oris, forearm, back of the hand, and palm. The results demonstrate that the highest power density of backscattering on the palm is around 1.02 mW/cm$^2$ and 1.42 mW/cm$^2$, as detected by the nearest sensor, which was 0.7 cm away from the 830 nm and 660 nm laser output center, respectively. The power density of backscattering light on a sensor, located 23 mm away from the laser output center, approaches zero.

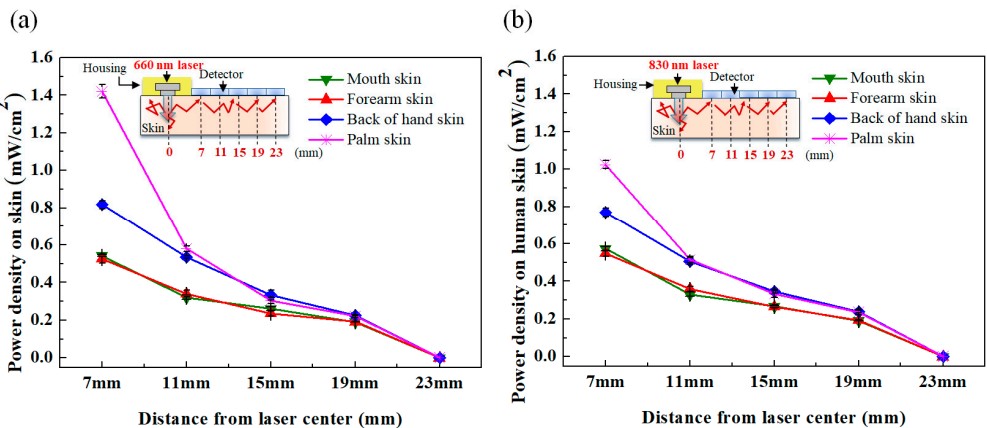

**Figure 4.** Backscattering of (**a**) 660 nm and (**b**) 830 nm lasers on different skin areas.

Furthermore, the power density distributions of 830 nm lasers with different focal points in the orbicularis oris were measured by nine detectors. The maximum power density on the attached gingiva of lasers with a 10 mm focal point was 1.89 mW/cm$^2$ (Figure 5a). Similarly, the maximum power density on the attached gingiva of lasers without a lens was 1.80 mW/cm$^2$. Meanwhile, the power densities of 830 nm lasers on the attached gingiva versus different focal points were analyzed (Figure 5b). In Figure 5c, the ratio of power densities on the attached gingiva at 4 mm, 8 mm, and 12 mm can be raised 1.81, 2.99, and 2.97-fold more than that laser without a lens, respectively (Figure 5c). Furthermore, the power density decay on the attached gingiva can be improved by 30.2% and 32.3% at 4 mm and 8 mm away from the laser center, respectively (Figure 5d).

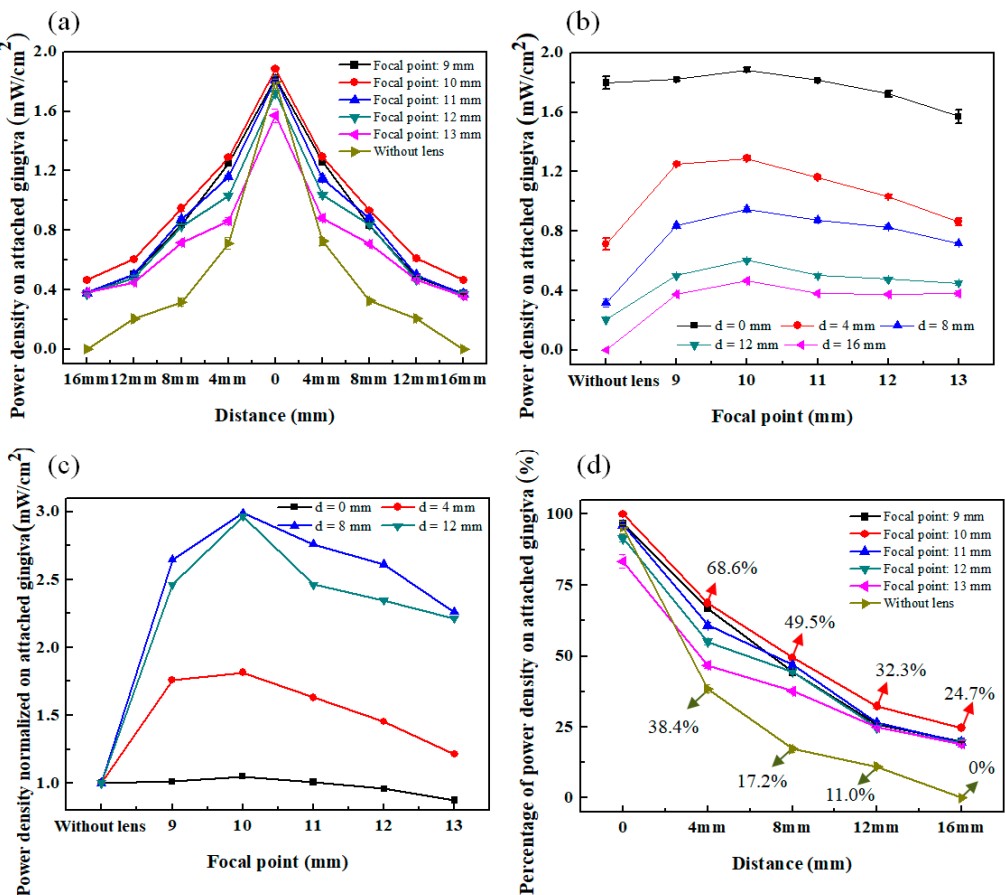

**Figure 5.** (**a**) Power densities distribution of the 830 nm laser penetrates the orbicularis oris with different focal points on the attached gingiva. (**b**) Power densities of the 830 nm laser on the attached gingiva versus different focal points. (**c**) Normalized power densities of the 830 nm laser on the attached gingiva versus different focal points. (**d**) Percentage of power densities decay on the attached gingiva from 0 mm to 16 mm.

## 4. Discussion

Skin is a turbid medium that causes the occurrence of multiple light scattering. Light scattering occurred when the laser penetrated the skin, especially in the dermis and subcutaneous layers (Figure 2c,f). In this work, 660 nm and 830 nm laser propagation in tissue was investigated. The backscattering of the 660 nm laser on the palm and back of the hand is higher than that of the 830 nm laser (Figure 4). However, the power density of skin is very small compared to the incident power of lasers (30 mW). On the other hand, different skin areas, such as under the beard or hair, could influence the power density of backscattering light; color and thickness of the skin may also have an influence. In this case, the skin colors of the palm and back of the hand are slightly higher than those of the orbicularis oris and forearm. Therefore, the power density of backscattering light on the palm and back of the hand is higher than that on the orbicularis oris and forearm [37,38]. When a 830 nm laser penetrated the skin, a slight difference of maximal power density was observed between lasers with a lens and without a lens (Figure 5a). In general, the maximal power density of an 830 nm laser on the target is approximately the same whether the lens is used or not. On the other hand, a higher power density on the peripheral tissues can be achieved by using an 830 nm laser with a 10 mm focal point in the orbicularis oris. The results show that the power densities on the peripheral tissues can be raised approximately three-fold at 8 mm to 12 mm from the 830 nm laser center (Figure 5c). Moreover, the power density decay of a 830 nm laser on the attached gingiva can be improved by 30.2% and 32.3% at 4 mm and 8 mm away from the output center, respectively (Figure 5d). The light at 660 nm

is more risky than 830 nm at the same dose on the orbicularis oris. The incident beam of the 830 nm laser can achieve better power density distribution in deeper tissue. On the contrary, the light 660 nm laser concentrates on shallow skin; thus, the dosage is a big issue. Using a high dosage of laser at 660 nm and focusing it on shallow tissue would cause pain and even damage of the tissue.

In our case, the best focal point for the 830 nm laser to penetrate the orbicularis oris is 10 mm. Based on our results, the incident beam of the NIR laser can be designed to achieve better power density distribution in the targeted tissue. In dentistry applications, the power densities on the peripheral tissues of attached gingiva can be improved when a 830 nm laser is used with an adequate focal point in the orbicularis oris, as shown in Figure 6. From our results, the power densities of the 830 nm laser with a 10 mm focal point can be achieved approximately 1 mW/cm$^2$ (0.95 mW/cm$^2$) at 8 mm from the laser center (Figure 5a). These results can be applied to traditional Chinese medicine (TCM) applications. When the LA ($\lambda$ = 830 nm, 30 mW) with a 10 mm focal point is applied to the skin with a thickness of 1 cm, the power density at range of 8 mm from the laser center can be calculated approximately 1 mW/cm$^2$. This means the maximal power density (1.89 mW/cm$^2$) is nearly two-fold compared to the surrounding tissues at 8 mm (0.95 mW/cm$^2$) from the laser center.

According to recent studies, if more mitochondria activity can be achieved by the laser with a uniform distribution [39], the uniformity of PBM treatment can improve the experimental outcomes. This study is very consistent with our other research about antibacterial efficacy of photodynamic therapy (PDT) [40]. The results of the study highlight the importance of the laser profile as a key parameter that determines the survival rate of *Actinobacillus actinomycetemcomitans* and *Streptococcus mutans* at the edge of the culture plate. Thus, the dose is an important parameter for PMB [39] and PDT [40].

Our experimental results show that the power density distribution on the target tissue can be increased by an 830 nm laser with a suitable lens. That is, an adequate focal point is key for NIR laser distribution in tissue, and the dose can be quantitatively calculated for healing time on the target and its surrounding tissues. For example, the healing time can be quantitatively calculated for nearly 9 min when the total dose (1 J/cm$^2$) of PBM is used in 1 cm thickness of tissue (1000 mJ/cm$^2$/1.89 mW/cm$^2$ $\approx$ 530 s $\approx$ 9 min). The exact dose of PBM therapy can increase the efficiency for clinical research and applications.

According to the Arndt–Schulz Law, a lower dose of PBM is considered for the positive effect in biological response [6–8]. However, the results should be reassessed. In a recent study, the photoresponse of isolated mitochondria in bovine liver irradiated by a 980 nm laser (7.69 to 107.69 J/cm$^2$) was investigated [41]. The results indicate that a 980 nm laser at 7.69 J/cm$^2$ (low energy density) causes drastic inhibition of ATP production. The optimal dose ranges from 61.54 to 84.62 J/cm$^2$. If the light dose is wrongly administered, the photoresponse could affect the mediators of the cells. Similarly, biphasic dose responses have also been observed in our clinical studies [4,5]. Thus, different doses of PBM therapy can create different effects in clinical applications. Repeated and more standardized studies are required to clarify the therapeutic strategies of PBM applications.

In this work, the optical characteristics of light propagation in skin were investigated. The power densities on the target and peripheral tissues were successfully demonstrated by a noninvasive sensing system. Our results are helpful for optimizing doses of PBM in different skin areas, especially LA applications.

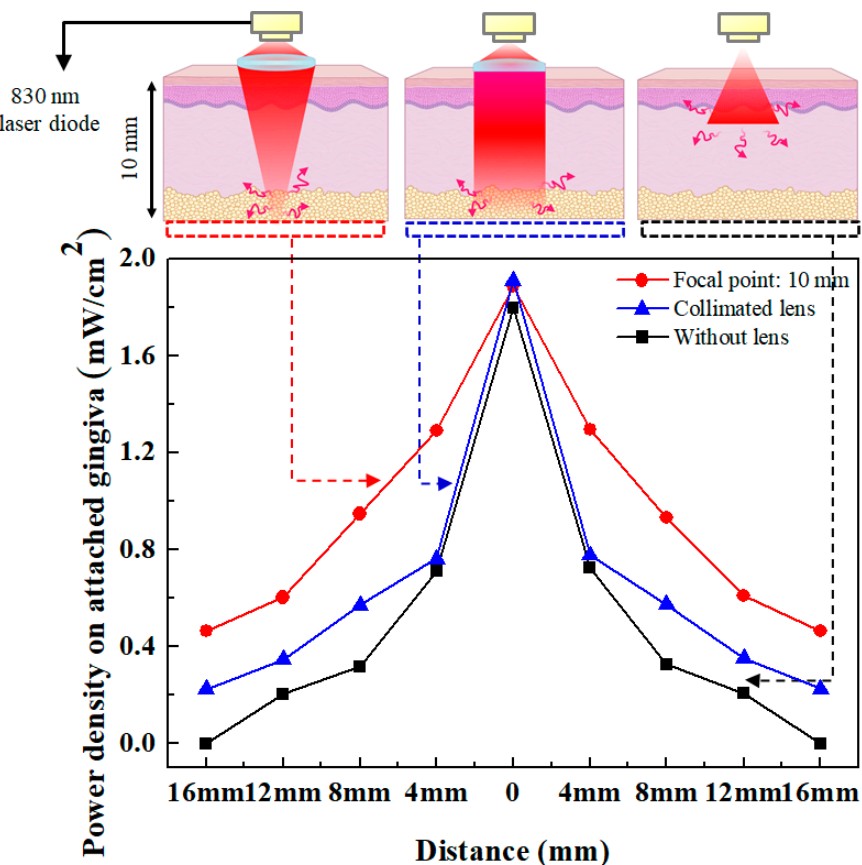

**Figure 6.** Power densities distribution of the 830 nm laser with a different laser beam penetrating the orbicularis oris. Schematics were created with BioRender.com.

## 5. Conclusions

In the present study, the power densities on the attached gingiva radiated with a variable focal point laser were investigated. The optimal focal point in the orbicular oris was found and the power densities on the surrounding tissues can be raised. Our present method reveals that, if the focal point of an NIR laser can be determined, the dose in the tissue can be defined. Furthermore, the focal point of an NIR laser should be focused in the tissue for safe LA applications. The method is useful for increasing the power density on the target and its surrounding tissue, especially for LA. According to TCM, acupoints are located at different depths of our body; thus, focusing lasers on the acupoints could increase the efficiency of PBM in clinical applications.

**Author Contributions:** Conceptualization and investigation, C.-T.S. and J.-H.W.; methodology and validation, C.-T.S. and J.-H.W.; writing—original draft preparation, C.-T.S.; writing—review and editing, F.-C.C. and J.-H.W.; simulation work, S.-H.M.; project administration, C.-T.S. and J.-H.W. All authors have read and agreed to the published version of the manuscript.

**Funding:** This research received no external funding.

**Informed Consent Statement:** Informed consent was obtained from a male case involved in the study.

**Data Availability Statement:** The data used to support the findings of this study are included within the article.

**Conflicts of Interest:** The authors declare no conflict of interest.

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
