# Peer review of "Optimization of Photobiomodulation Dose in Biological Tissue by Adjusting the Focal Point of Lens"

_photonics, doi:10.3390/photonics9050350_

Round 1

Reviewer 1 Report

Dear Authors,
your data support PBM consistency, which has to be one of the main subjects for PBM research. 
The work needs only minor revisions and clarification according to recent literature evidence. 

Indeed, the Arndt-Schulz Law supported much experimental evidence in past. However, a recent paper* on the primary target of PBM clearly shows window effects that could allow new interpretations for PBM events. This evidence supports the needing for irradiation therapy that has to be consistent, standardised and reproducible.   
*Andrea Amaroli, Claudio Pasquale, Angelina Zekiy, Anatoliy Utyuzh, Stefano Benedicenti, Antonio Signore, Silvia Ravera, "Photobiomodulation and Oxidative Stress: 980 nm Diode Laser Light Regulates Mitochondrial Activity and Reactive Oxygen Species Production", Oxidative Medicine and Cellular Longevity, vol. 2021, Article ID 6626286, 11 pages, 2021.

Recently, authors** showed that irradiating probes have different beam energy distribution features. These differences significantly affected mitochondria activity with respect to their position in the treatment spot size. 
Please, compare and discuss your results and the recent results in the discussion section. 
**Amaroli A, Arany P, Pasquale C, Benedicenti S, Bosco A, Ravera S. Improving Consistency of Photobiomodulation Therapy: A Novel Flat-Top Beam Hand-Piece versus Standard Gaussian Probes on Mitochondrial Activity. Int J Mol Sci. 2021 Jul 21;22(15):7788

Author Response

We really appreciate the reviewer for his/her very valuable and constructive comments on our paper.

Point (1) Indeed, the Arndt-Schulz Law supported much experimental evidence in past. However, a recent paper* on the primary target of PBM clearly shows window effects that could allow new interpretations for PBM events. This evidence supports the needing for irradiation therapy that has to be consistent, standardised and reproducible.

*Andrea Amaroli, Claudio Pasquale, Angelina Zekiy, Anatoliy Utyuzh, Stefano Benedicenti, Antonio Signore, Silvia Ravera, "Photobiomodulation and Oxidative Stress: 980 nm Diode Laser Light Regulates Mitochondrial Activity and Reactive Oxygen Species Production", Oxidative Medicine and Cellular Longevity, vol. 2021, Article ID 6626286, 11 pages, 2021.

Response 1: The below paragraph has been added in Discussion and the reference also has been cited in article, started from page 8, line 217-225.

According to the Arndt-Schulz Law, lower dose of PBM is considered for the positive effect in biological response [6-8]. However, the results should be reassessed. In recent study, the photoresponse of bovine liver isolated mitochondria irradiated by 980 nm laser (7.69 to 107.69 J/cm2) were investigated [41]. The results indicated 980 nm laser at 7.69 J/cm2 (low energy density) cause drastic inhibition of ATP production. The optimal dose is ranged from 61.54 to 84.62 J/cm2. If the light dose wrongly administered, the photoresponse could affect mediators of cells. Similarly, biphasic dose response also has been observed in our clinic studies [4, 5]. Thus, different dose of PBM therapy can create different effects in clinical applications. Repeated and more standardized studies are required to clarify the therapeutic strategies of PBM applications.

Point (2) Recently, authors** showed that irradiating probes have different beam energy distribution features. These differences significantly affected mitochondria activity with respect to their position in the treatment spot size.

Please, compare and discuss your results and the recent results in the discussion section.

**Amaroli A, Arany P, Pasquale C, Benedicenti S, Bosco A, Ravera S. Improving Consistency of Photobiomodulation Therapy: A Novel Flat-Top Beam Hand-Piece versus Standard Gaussian Probes on Mitochondrial Activity. Int J Mol Sci. 2021 Jul 21;22(15):7788

Response 2: The paragraph has been added in Discussion as below, started from page 8, line 203-209.

In recent study, more mitochondria activity can be achieved by the laser with a uniform distribution [39], the uniformity of PBM treatment improve the experimental outcomes. This study is very consistent with our other research about antibacterial efficacy of photodynamic therapy (PDT) [40]. The results of the study highlighted the importance of laser profile as a key parameter that determines the survival rate of Actinobacillus actinomycetemcomitans and Streptococcus mutans at the edge of the culture plate. Thus, the dose is an important parameter for PMB [39] and PDT [40].

Reviewer 2 Report

The near infrared spectroscopy was widely used in field such as agriculture, medicine and so on. I think this article is a good publication that expands the range of NIR usage for medical dentistry. Good scientific work on light absorption and light scattering. I wrote few question and comments below.

Line 48-51: The well transmitted wavelength differs depending on the measurement sample. Was such difference of light penetration due to the measurement sample cell size?

Figure 2: Regarding the measurement in the mouth, the thickness of tooth crown or light path length was differed depending on the measurement position. Was the measurement value corrected?

Figure 4: Backscattering of 660 nm and 830 nm was not detected at 23 mm (approached 0 value). Why was the difference due to wavelength not detected ?

Line 180:The skin colors affected the light penetration. In particular, using 660nm, was there any effect on the repeatability because of the individual difference.

Conclusions: Could it actually be used for diagnosis? Is it possible to detect the oral disease?

Author Response

Point 1: Line 48-51: The well transmitted wavelength differs depending on the measurement sample. Was such difference of light penetration due to the measurement sample cell size?

Reply: “The light penetration depends on wavelength and also the sample used in the experiment. The well transmitted wavelength would be different for human ex vivo tissues [12] and animal tissues [13].” These sentences are added in “Introduction”, started from page 2, line 52-54.

Point 2: Figure 2: Regarding the measurement in the mouth, the thickness of tooth crown or light path length was differed depending on the measurement position. Was the measurement value corrected?

Reply: In the experiment, the real power densities of 830 nm laser after penetrating the orbicularis oris on the attached gingiva were measured. A flexible facial fixture appliance provides a stable position for laser source (including laser device and lens device) and the retainer provides a stable position for sensing system detection. A laser was fixed in the center of a flexible facial fixture appliance. We just changed the focal point of the lens, or removed the lens. The thickness of orbicularis oris and the measurement position are fixed. Thus the measurement values are reliable.

Point 3:Figure 4: Backscattering of 660 nm and 830 nm was not detected at 23 mm (approached 0 value). Why was the difference due to wavelength not detected?

Reply: When the light propagation achieved 23 mm away from the laser center, the power densities is too small to be detected. Both of 660 nm and 830 nm laser demonstrated the same results for the light propagation decay on skin.

Point 4: Line 180: The skin colors affected the light penetration. In particular, using 660nm, was there any effect on the repeatability because of the individual difference.

Reply: In the experiment, an Asian case was included in our study. From the experimental results, the different skin areas can influence the power density of backscattering light, such as color, thickness, beard or hair. The individual difference can affect the optical characteristics, including skin color, BMI, different age groups and people with different living habits. The individual difference is an important issue. It is worth further study in the future.

Point 5: Conclusions: Could it actually be used for diagnosis? Is it possible to detect the oral disease?

Reply: It is possible to detect the oral diseases. Because different oral diseases have different tissue situations. The refractive index in different tissue layers would be different. Thus, the light propagation would be changed. The experiment protocol and the sensing area should be redesigned for further study.

Reviewer 3 Report

In this article, the authors realized modulation of laser power densities in biological tissues by adjusting the focal point of the lens, which is promising for Traditional Chinese Medicine applications, especially noninvasive laser acupuncture. In the experiment, the authors attached the flexible facial fixture appliance (including laser device, lens device, and sensing system) to a man’s gingiva to investigate 830 nm and 660 nm laser propagation in tissues (epidermis layer, dermis layer, subcutaneous layer). The results showed that the power densities on the peripheral tissues can be raised nearly 3-fold at 8 mm to 12 mm from the 830 laser center, while the power density decay can be improved by 32.3% at 8 mm from the laser center. In addition, the best focal point for 830 nm laser penetrated orbicularis oris is measured to be 10 mm. In conclusion, this paper indicated that a collimated laser under a suitable lens can improve the power densities efficiently without severe damage to the skin. Overall, this article is a good-written one with high-quality diagrams and modulation, concise presentation, logical analysis, and discussion. I would only provide several minor suggestions as followed:
1. From line 34 to line 40, the authors give a brief introduction of the dose of PBM. I would suggest putting these statements after the first sentence “Photobiomodulation has been published in many clinical applications” to make this part logically connected.
2. I suggest adding a reference in line 50 “in animal tissues, a minor attenuation on the rat skin, pig fat and pig muscle with 904 nm laser were measured”.
3. The authors conducted the experiment using 660 nm and 830 nm lasers. But I don’t understand why you simulated the optical characteristics of 660 nm and 825 nm lasers. The same confusion exists in Figure 2.
4. I suggest putting Figure 1 in the first paragraph of materials and methods.
5. I would recommend unifying the coordinate scale of (a) and (b) to make it easier for readers to compare.
6. In the discussion part, “the backscattering of 660 nm and 830 nm laser propagation ... 830 nm laser” in line 176, and “the backscattering of 660 nm laser radiated ... 830 nm laser” in line 178 are the same. I would suggest deleting one of them. 
7. I’m wondering how the dose can be quantitatively calculated for healing time on the target and its surrounding tissues as mentioned in line 202. I suggest an added explanation or corresponding reference. 
Consideration of all, this paper is well-written and suitable for the scope of our journal.

Author Response

Point (1): From line 34 to line 40, the authors give a brief introduction of the dose of PBM. I would suggest putting these statements after the first sentence “Photobiomodulation has been published in many clinical applications” to make this part logically connected.

Reply: This paragraph has been revised in the Introduction as below:

“Light wavelengths (nm), power density (W/cm2), energy density (J/cm2), operation frequency (Hz), spot size (cm2) and penetration depth are key parameters for the PBM applications. PBM has been published in many clinical applications. In PBM, the dose is an important parameter for biological photoresponse. … The suitable dose on the target can increase the efficiency of PBM clinical applications.” This paragraph is shown in “Introduction”, started from page 1, line 36.

Point (2): I suggest adding a reference in line 50 “in animal tissues, a minor attenuation on the rat skin, pig fat and pig muscle with 904 nm laser were measured”.
Reply: The reference has been added as bellow:

In animal tissues, a minor attenuation on the rat skin, pig fat and pig muscle with 904 nm laser were measured [13]. This sentence is shown in “Introduction”, started from page 2, line 50.

Point (3): The authors conducted the experiment using 660 nm and 830 nm lasers. But I don’t understand why you simulated the optical characteristics of 660 nm and 825 nm lasers. The same confusion exists in Figure 2.
Reply: From published paper, the simulation of 825 nm lasers in skin are more complete than that of 830 nm laser. The optical characteristics in different wavelengths have been investigated in previous study [1]. The absorption coefficient, reduced scattering coefficient and optical penetration depth between 825 nm and 830 nm laser in human skin are very close. And the photoresponse for ATP synthase between 825 nm and 830 nm laser are very close too [2]. In our recent study, the optical simulation result of 825 nm laser in skin model was similar to 830 nm laser on real skin [3]. Therefore, the optical characteristics of 660 nm and 825 nm lasers were used in the present study.

[1]   Bashkatov, A.N.; Genina, E.A.; Kochubey, V.I.; Tuchin, V.V. Optical properties of human skin, subcutaneous and mucous tissues in the wavelength range from 400 to 2000 nm. J Phys D: Appl Phys. 2005, 38, 2543–2555.

[2]   Desmet, K.D.; Paz, D.A.; Corry, J.J.; Eells, J.T.; Wong-Riley M.T.T.; Henry, M.M.; Buchmann, E.V.; Connelly, M.P.; Dovi, J.V.; Liang, H.L.; et al. Clinical and experimental applications of NIR-LED photobiomodulation. Photomed. Laser Surg. 2006, 24, 121–128.

[3] Su, C.T.; Chen, C.M.; Chen, C.C.; Wu, J.H. Dose Analysis of Photobiomodulation Therapy in Stomatology. Evid-Based Complement. Alternat. Med. 2020, 2020, 1–12.

Point (4): I suggest putting Figure 1 in the first paragraph of materials and methods.
Reply: The Figure 1 has been moved to the first paragraph in “Material and Methods”.

Point (5): I would recommend unifying the coordinate scale of (a) and (b) to make it easier for readers to compare.

Reply: The coordinate scale of (a) and (b) in Figure 4 have been unified.

Point (6): In the discussion part, “the backscattering of 660 nm and 830 nm laser propagation ... 830 nm laser” in line 176, and “the backscattering of 660 nm laser radiated ... 830 nm laser” in line 178 are the same. I would suggest deleting one of them. 

Reply: The sentence “the backscattering of 660 nm laser radiated ... 830 nm laser” in line 178 has been deleted.

Point (7): I’m wondering how the dose can be quantitatively calculated for healing time on the target and its surrounding tissues as mentioned in line 202. I suggest an added explanation or corresponding reference. 

Reply: The light dose is the product of power density and duration time. In our study, the power density distribution of 830 nm laser in 1 cm thickness of tissue can be measured. “For example, the healing time can be quantitatively calculated for nearly 9 minutes when the total dose (1 J/cm2) of PBM is used in 1 cm thickness of tissue (1000 mJ/cm2∕ 1.89 mW/cm2 ≈ 530 second ≈ 9 minutes). The exact dose of PBM therapy can increase the efficiency for clinical research and applications.” The dose calculation has been added in “Discussion”, started from page 8, line 213-216.